# OpenReview forum: "Compositional Risk Minimization"
_ICLR.cc/2025/Conference — Submitted to ICLR 2025_

### Official Review · Reviewer_MuZN · 2024-10-28

**Soundness:** 3
**Presentation:** 4
**Contribution:** 3
**Rating:** 6
**Confidence:** 4

**Summary:**

This paper focus on tackling the challenge of  subpopulation shift through Compositional risk minimization, where the theory of discriminative compositional shifts is established, and a practical algorithm (compositional risk minimization, CRM) is also given. Overall, the theory are technically sound  and the experimental results demonstrates that CRM is . A good paper but require more specifications.

**Strengths:**

1. **Novel Approach**: The paper introduces a novel approach based on flexible additive energy models for compositionality in discriminative tasks, which can provably extrapolate to novel attribute combinations.
2. **Theoretical Grounding**: The approach is theoretically grounded, providing guarantees for generalization under compositional shifts, which is crucial for robust machine learning models.
3. **Comprehensive Evaluation**:  Extensive empirical studies and experiments to demonstrate the impact of different choices and validate the effectiveness of the proposed method.

**Weaknesses:**

1. The Compositional Risk Minimization (CRM) framework lacks explicit constraints on feature representations, which is widely recognized as crucial for generalization under sub-population shifts [1]. While CRM effectively models compositional shifts through additive energy distributions, it does not incorporate mechanisms to ensure the learned features are robust and semantically meaningful across different attribute combinations. Could the author make a further discussion on this problem?

    [1] Liu et al., Causal Triplet: An Open Challenge for Intervention-centric Causal Representation Learning, 2023

2. For the implementation of Algorithm 1, how to pre-specifies the spurious attribute $a$?

3. The current experimental results are somewhat lacking of conviction. More advanced DG algorithms, e.g. IRM, VREx, Fishr, SelfReg, are recommended to be included in the experiments.

4. The sample complexity $s$ w.r.t the dimension $d$ should be verified through simulation studies.

**Questions:**

1. I am a little confused of the assumption 1: $p(x|z)=q(x|z)$，i.e.  the data generation mechanism from attributes to the inputs remains invariant, which is the main claim in the paper,  hence the distribution shift is essentially caused by $p(z)\neq q(z)$, which further implies that $p(x)\neq q(x)$, due to  $p(x)= \int p(x|z)p(z)dz$, and $q(x)=\int q(x|z)q(z)dz$. So, the covariate shift is caused by the marginal differences on the attributes. This seems contradict to traditional DG.

2. For sample complexity $s\geq 8cd\log(d/2)$,  $c$ lacks of specifications?

---

> ### Author Response · Authors · 2024-11-20
> **Author Rebuttal: Part 1/2**
>
> We thank the reviewer for their insightful and constructive feedback. Please see our responses below and let us know if you have further questions.
>
> &nbsp;
>
> ## **Constraints on Feature Representations**
> The additive energy distributions, which are at the foundation of our approach, are inspired by the independent mechanisms principle. These additive energies appear in our classifier and constrain the feature representations. It is this inductive bias that leads to improved robustness as demonstrated in our experiments. The energy components in our approach are semantically meaningful and "disentangled",  as each component corresponds to different attribute. Building disentangled and semantically meaningful representations based on additive energy distribution is a future work we are excited about.
>
> &nbsp;
>
> ## **Pre-specifying Attribute $a$**
> To pre-specify the attribute $a$, we use whatever is standard for that dataset in the literature. For instance, in the Waterbirds dataset it is common to choose $y$ to be the bird and $a$ to be the background. However, our training is independent of which attribute is spurious as we learn to predict the joint distribution over all the attributes.
>
> &nbsp;
>
> ## **Additional Baselines**
> Based on your suggestion, we have added IRM and VREx, and we are in the process of benchmarking Fishr as well. The results are presented below, with more detailed results in Table 14, Appendix E.2 for the datasets Waterbirds, CelebA, MetaShift, NICO++, and MultiNLI. We will soon add results for CivilComments as well, since this dataset is much larger than the others (16 different compositional shift scenarios) it will take longer to finish experiments on them.
>
> Our primary comparison baseline has been with GroupDRO, as it is the most effective method for addressing subpopulation shifts, hence it was the main focus of our work. We also included baselines like LA (Logit Adjustment) that share conceptual similarities with our approach. Baselines such as IRM and VREx were initially excluded because they are less suited for tackling subpopulation shifts, we have now incorporated them into our evaluation.
>
>
>
>    | Dataset    | Method | Average Acc | Balanced Acc | Worst Group Acc |
>    |------------|--------|-------------|--------------|-----------------|
>    | Waterbirds | IRM    | 73.6 (0.8)  | 70.4 (0.3)   | 28.7 (2.2)      |
>    |            | VREx   | 81.0 (0.6)  | 80.0 (0.5)   | 45.6 (1.1)      |
>    |            | **CRM**    | **87.1 (0.7)**  | **87.8 (0.1)**   | **78.7 (1.0)**      |
>    | CelebA     | IRM    | 80.4 (1.3)  | 76.7 (1.1)   | 40.1 (2.4)      |
>    |            | VREx   | 86.2 (0.3)  | 82.8 (0.5)   | 49.2 (2.1)      |
>    |            | **CRM**    | **91.1 (0.2)**  | **89.2 (0.0)**   | **81.8 (0.5)**      |
>    | MetaShift  | IRM    | 83.7 (0.3)  | 80.3 (0.4)   | 55.8 (1.0)      |
>    |            | VREx   | 84.9 (0.4)  | 81.7 (0.3)   | 59.9 (0.2)      |
>    |            | **CRM**    | **87.6 (0.3)**  | **84.7 (0.2)**   | **73.4 (0.4)**      |
>    | NICO++     | IRM    | 64.0 (0.6)  | 62.7 (0.3)   | 0.0 (0.0)       |
>    |            | VREx   | 86.0 (0.0)  | 86.0 (0.0)   | 37.3 (4.3)      |
>    |            | **CRM**    | **84.7 (0.3)**  | **84.7 (0.3)**   | **40.3 (4.3)**      |
>   | MultiNLI | IRM    | 65.7 (0.1) | 63.7 (0.4) | 8.1 (0.8) |
>  |                | VREx   | 69.0 (0.0) | 68.8 (0.2) | 4.1 (0.3) |
>  |                | **CRM**    | **74.6 (0.5)** | **76.2 (0.6)** | **57.7 (3.0)** |
>
> We find that CRM **significantly outperforms** IRM and VREx w.r.t worst group accuracy. In fact, CRM also obtains better performance than IRM and VREx w.r.t average accuracy and group balanced accuracy as well.
>
> &nbsp;
>
>
> ##  **Invariance of  p(x|z)**
> There are two types of data generation mechanisms in the literature on causally inspired domain generalization. In the former one, the assumption is the invariance of $p(y|z)$ and $p(z)$ varies [1] (with $z=x$ in the case of covariate shift [5]). In the latter one, the assumption is the invariance of $p(x|z)$ and $p(z)$ varies [2][3] (with $z=y$ in the case of label shift [4]). Our assumptions are closer to the latter line of work.
>
> &nbsp;
>
> ### **References**
>
>
> [1] Arjovsky, Martin, et al. "Invariant risk minimization." arXiv preprint arXiv:1907.02893 (2019).
>
> [2] Rosenfeld, Elan, Pradeep Ravikumar, and Andrej Risteski. "The risks of invariant risk minimization." arXiv preprint arXiv:2010.05761 (2020).
>
> [3] Wang, Haoxiang, et al. "Invariant-Feature Subspace Recovery: A New Class of Provable Domain Generalization Algorithms." arXiv preprint arXiv:2311.00966 (2023).
>
> [4] Garg, Saurabh, et al. "A unified view of label shift estimation." Advances in Neural Information Processing Systems 33 (2020): 3290-3300.
>
> [5] Bickel, Steffen, Michael Brückner, and Tobias Scheffer. "Discriminative learning under covariate shift." Journal of Machine Learning Research 10.9 (2009).

---

> ### Author Response · Authors · 2024-11-23
> **Author Rebuttal: Part 2/2**
>
> ## **What is c?**
> $c$ controls the probability of success given by $1-1/c$.
>
>
> &nbsp;
>
> ## **Complexity Analysis as a Function of $d$**
>
> We really appreciate this question and indeed believe its an interesting analysis. We did experiments on a synthetic dataset to show the group complexity $s$, i.e., number of groups as a function of $d$.
>
> We consider the case of two attributes $z= (z_1, z_2)$, each can take $d$ possible categorical values. We sample data from the following (additive) energy function.
> $$E(x, z)= || x - \mu(z_1) ||^2 + || x - \mu(z_2) ||^2 $$
>
> where $x, \mu(z_1), \mu(z_2) \in \mathbb{R}^{n}$.  We fix the data dimension as $n=100$ and vary the total categories per attribute $d$ in the following range $[10, 20, 30, 40, 50]$. Then for each scenario $(n, d)$ we analyze how the performance of CRM degrades as we discard more groups from the training dataset. Please refer to Appendix E.3 for more details regarding the setup.
>
> Figure 6 (Appendix E.3) presents the results of our analysis. We find that CRM trained with $20$ % of the total groups still shows good generalization ($\sim 90\%$ test accuracy), and the drop in test accuracy as compared to the oracle case of no groups dropped is within $10$ % .

---

> > ### Comment · Reviewer_MuZN · 2024-11-26
> > **response to the rebuttal**
> >
> > Thank you for the response.  Most of my concerns are addressed, and I would like to maintain my score.

---

> > > ### Author Response · Authors · 2024-11-26
> > >
> > > We sincerely appreciate the time and effort you have dedicated to reviewing our manuscript. We are happy to see that our rebuttal has addressed your concerns. If there are any additional questions or concerns where further clarification is needed, please do not hesitate to reach out, and kindly consider increasing your score.
> > >
> > > In particular, we would like to invite you to reconsider whether the significance of our **theoretical contribution is appropriately weighted in your assessment**. Our method is the first to offer *provable guarantees* for extrapolating under compositional shifts, which also stands the test of experimentation on real benchmark datasets, as CRM systematically outperforms all baselines under such shifts.

---

### Official Review · Reviewer_Rmiq · 2024-11-01

**Soundness:** 3
**Presentation:** 3
**Contribution:** 3
**Rating:** 6
**Confidence:** 4

**Summary:**

This paper addresses a challenging scenario involving compositional shifts, where certain combinations of attributes are not present in the training distribution but appear in the test distribution. To tackle this issue, the authors propose a Compositional Risk Minimization (CRM) framework, which incorporates an additive energy classifier to predict multiple attributes. Furthermore, the framework adapts this classifier to effectively handle compositional shifts. Empirical studies demonstrate the effectiveness of the proposed CRM approach.

**Strengths:**

This paper explores a significant and under-researched area related to compositional shifts, which has been largely overlooked in the existing literature.

The study is quite rigorous and comprehensive. Notably, it introduces the proposed approach through a running example involving two attributes, which makes the contributions and motivations clear and accessible. Additionally, the formulation of the Compositional Risk Minimization (CRM) framework offers valuable insights for future research.

**Weaknesses:**

One significant concern pertains to the attribute distributions during compositional shifts, which are frequently encountered in real-world applications. For instance, the distribution may shift from uniform to long-tailed, or it may be long-tailed from the outset. In this context, is it still effective to assume the test group prior $q(z)$ to be uniform? I encourage the authors to consider and discuss this issue in their paper.

The efficiency of the proposed CRM approach should also be examined, particularly in the context of multiple attribute abstention cases.

**Questions:**

The authors should address all my concerns presented in the Weakness part.

---

> ### Author Response · Authors · 2024-11-20
> **Author Rebuttal**
>
> Thank you for your insightful and constructive feedback. Please see our responses below, and reconsider if we addressed your concerns. Also let us know if you have further questions.
>
> &nbsp;
>
> ## **Uniform vs Imbalanced Prior**
>
> This is indeed an important point to better highlight and discuss. Thank you for suggesting this.
> Note that we had already briefly explored the use of different test priors in Table 10 of the Appendix.
> We are renaming that section of the appendix into "Choice of test prior and importance of extrapolated bias" and adding the following expanded discussion regarding the choice of test prior:
>
> Our algorithm allows the flexibility at test time to choose the prior over test groups as we see fit. That choice should be informed by what we might know or guess about the distribution over test groups, as well as what the metric of interest is. If we assume we can know nothing about the test group distribution, and we care about robust metrics invariant to changes in that distribution, such as balanced-group accuracy or worst-group-accuracy (WGA), then it makes sense to use a uniform prior over groups. This is what we did in most of our experiments.
> On the other extreme, if we can estimate the test group distribution and what we care about is average test accuracy, then we should use that estimate (which we call empirical prior) as the test prior. We explore these alternatives and show the results in Table 10.  As expected from the theory, the empirical prior achieves better average test accuracy, while the uniform prior performs better in terms of worst group accuracy.
>
> Other choices of test prior are possible, depending on what we know and care about. For e.g. if we know some attribute combinations are impossible at test time, we can set their prior probability to 0, while e.g. keeping it uniform over all other possible combinations. Or if we assume that the marginal test distribution of each attribute will be close to its marginal train distributions, we can estimate these marginals on the training set and define an independent test prior as their product. Exploration of the practical usefulness of such alternatives is left as future work.
>
>
> &nbsp;
>
> ## **Multi-Attribute Experiments**
> Following your suggestion to include empirical results with more attributes we have carried out a new experiment with CelebA using three attributes. This new results table (Table 13 in Appendix E.1) is also reproduced in the general response above, for convenience. CRM continues to be the more robust approach.
>
> | Method | Average Acc | Balanced Acc | Worst Group Acc | WGA (No Groups Dropped) |
> |--------|-------------|--------------|-----------------|-------------------------|
> | ERM    | 94.1 (0.1)  | 77.4 (0.3)   | 21.6 (0.6)      | 19.0 (0.0)              |
> | G-DRO  | 91.2 (0.5)  | 87.3 (0.3)   | 61.0 (2.1)      | 66.7 (2.3)              |
> | LC     | 92.8 (0.0)  | 89.8 (0.1)   | 75.0 (1.3)      | 77.0 (2.0)              |
> | sLA    | 93.0 (0.1)  | 89.7 (0.1)   | 76.0 (0.7)      | 77.0 (4.0)              |
> | **CRM**    | **92.3 (0.2)**  | **91.6 (0.0)**   | **84.0 (0.3)**      | **86.3 (1.2)**  |

---

> > ### Comment · Reviewer_Rmiq · 2024-11-25
> >
> > I appreciate the authors' effort in rebuttal. Most of my concerns have been addressed.

---

> > > ### Author Response · Authors · 2024-11-26
> > >
> > > We sincerely appreciate the time and effort you have dedicated to reviewing our manuscript. We are happy to see that our rebuttal has addressed your concerns. If there are any additional questions or concerns where further clarification is needed, please do not hesitate to reach out, and kindly consider increasing your score.
> > >
> > > In particular, we would like to invite you to reconsider whether the significance of our theoretical contribution is appropriately weighted in your assessment. Our method is the first to offer provable guarantees for extrapolating under compositional shifts, which also stands the test of experimentation on real benchmark datasets, as CRM systematically outperforms all baselines under such shifts.

---

### Official Review · Reviewer_x1T8 · 2024-11-04

**Soundness:** 3
**Presentation:** 3
**Contribution:** 3
**Rating:** 6
**Confidence:** 2

**Summary:**

This paper addresses the compositional shifts, a hard type of sub-population shifts, and proposes compositional risk minimization. The method is well-motivated and some theoretical analyses are provided. Results on the sub-population shift benchmark are shown to support the proposed method.

**Strengths:**

- The compositional risk minimization method is reasonable and well-motivated.
- The formulation of the compositional shift setting provides a foundation for further research.
- Experimental results are good to support the method.

**Weaknesses:**

Based on the theoretical analysis and experimental results, my main concerns are as follows:

1. **Experimental Setting**: The analysis and proposed algorithm are designed to handle multiple attributes, with the theoretical advantages being most relevant for this multi-attribute context. However, the experiments are limited to only two attributes. I suggest that the authors include empirical results with multiple attributes to better align with the theoretical analysis.

2. **Results**: While the CRM results outperform the baselines in this study, I check the original benchmark paper [1] and find that CRM’s performance does not consistently exceed that of the baselines. Furthermore, the authors only report accuracy, which doesn’t fully account for group imbalances. Reporting Macro-F1 scores would provide a more comprehensive evaluation.

3. **“Disjoint” Setting**: I understand that the paper’s chosen disjoint setting adds a level of complexity. However, in real-world scenarios, it is often feasible to obtain a small number of samples for different attribute combinations (particularly with only two attributes, as in these experiments). The proposed method should also be evaluated in traditional settings where all attribute combinations have some representation. This would confirm that the method performs well without requiring group-dropping.

**Questions:**

Please refer to weaknesses.

---

> ### Author Response · Authors · 2024-11-20
> **Author Rebuttal**
>
> Thank you for your insightful and constructive feedback. Please see our responses below, and reconsider if we addressed your concerns. Also, let us know if you have further questions.
>
> &nbsp;
>
> ## **Multi-Attribute Experiments**
> Following your suggestion to include empirical results with more attributes we have carried out a new experiment with CelebA using three attributes. This new results table (Table 13 in Appendix E.1) is reproduced below for convenience. CRM continues to be the more robust approach.
>
> | Method | Average Acc | Balanced Acc | Worst Group Acc | WGA (No Groups Dropped) |
> |--------|-------------|--------------|-----------------|-------------------------|
> | ERM    | 94.1 (0.1)  | 77.4 (0.3)   | 21.6 (0.6)      | 19.0 (0.0)              |
> | G-DRO  | 91.2 (0.5)  | 87.3 (0.3)   | 61.0 (2.1)      | 66.7 (2.3)              |
> | LC     | 92.8 (0.0)  | 89.8 (0.1)   | 75.0 (1.3)      | 77.0 (2.0)              |
> | sLA    | 93.0 (0.1)  | 89.7 (0.1)   | 76.0 (0.7)      | 77.0 (4.0)              |
> | **CRM**    | **92.3 (0.2)**  | **91.6 (0.0)**   | **84.0 (0.3)**      | **86.3 (1.2)**  |
>
> &nbsp;
>
> ## **Evaluation in the Traditional Setting**
> In the paper, we had already carried out a comparison in the traditional setting where all attribute combinations have some representation. These were presented in the rightmost column in Table 1, "WGA (no groups dropped)", as well as in Table 11 in Appendix which contains additional metrics. CRM remains competitive with the baselines in this scenario.
> This confirms that the method performs well also without requiring group-dropping.
>
> &nbsp;
>
> ## **Results: Baselines, Metrics, and Macro-F1 scores**
>
> Could you please clarify the work you had in mind as the "original benchmark paper [1]", as you did not provide the actual citation. Baseline performances therein may have been obtained in differing setups (e.g. without fine-tuning), and/or with different hyper-parameter selection strategies. Also, they were most likely not dropping groups, which is the more challenging setting we focus on.
>
> Regarding metrics that account for group imbalances, while average test accuracy does not, we reported group balanced accuracies in Appendix D.1 (Tables 5, 6, 7, 8, 9 containing the detailed results over all of the different discarded group scenarios). Group-balanced accuracy accounts for group imbalances as it essentially considers a uniform distribution over groups (see Appendix C.2 for details about the metrics). We had chosen to focus the main paper table on Worst Group Accuracy (WGA) -- which is also invariant under test-group rebalancing -- and is typically the metric of choice for robustness under subpopulation shifts. To not overload the main paper table we did not include balanced group accuracy therein.
>
> Following your suggestion, we have computed macro F1 scores and added these results in Table 14, Appendix E.2. The F1 scores are computed per group and then averaged across the groups. Macro F1 is akin to the group-balanced accuracy metric in our paper, except that it computes F1 score instead of accuracy per group. The results show that CRM is either better than or competitive with baselines w.r.t both macro F1 and group balanced accuracy.
>
> To summarize these findings, we also compute rankings of the method w.r.t a test metric and then report the average ranking across the different datasets (and their corresponding compositional shift scenarios). We share the average rank of each method below, as well as in Table 15, Appendix E.2
>
>    | Method | Average Acc | Balanced Acc | Macro F1 | Worst Group Acc |
> |---------------------|----------|----------|----------|----------|
> | ERM                | 3.53     | 3.97     | 3.63     | 4.73     |
> | G-DRO              | 3.48     | 3.19     | 2.74     | 3.83     |
> | LC                 | 1.98     | 1.93     | 1.93     | 2.91     |
> | sLA                | 1.99     | 2.10     | 2.02     | 3.26     |
> | IRM                | 4.96     | 5.06     | 4.45     | 5.68     |
> | VREx               | 3.45     | 3.43     | 3.28     | 4.74     |
> | CRM (Emp Prior)    | 1.61     | 3.00     | 2.57     | 3.61     |
> | **CRM**                | **2.56**     | **1.86**     | **1.60**     | **1.95**     |
>
> CRM obtains the best rank (lower the better) w.r.t balanced accuracy, macro F1, and worst group accuracy. Note that average accuracy is not the best indicator of generalization under spurious correlations. However, we can adapt CRM to utilize the empirical prior on test distribution to improve its average accuracy, as shown before in ablation studies with CRM (Table 10, Appendix D.2). In fact, as shown above, CRM with empirical prior obtains the best rank w.r.t average accuracy (at the cost of other metrics).

---

> > ### Comment · Reviewer_x1T8 · 2024-11-25
> >
> > Thank you for your rebuttal. I still have some questions: (1) Why the authors did not compare with Mixup (or C-Mixup), which is a really strong baseline in this setting; (2) Why only report the average rankings of different methods? A method with a better average ranking does not indicate it has a better overall performance, since it may be super bad in some cases. Given these concerns, I would like to maintain my score.

---

> ### Author Response · Authors · 2024-11-26
> **Regarding Unresolved Concerns**
>
> We thank the reviewer for their response to our rebuttal. Please take our responses below into consideration as we believe it should address your unresolved concerns.
>
> ## Mixup Baseline
>
> We are actively working on your suggestion to add the Mixup baseline. Below, we've provided results with Mixup for the Waterbirds, CelebA, MetaShift, and NICO++ datasets; please check Table 16 in Appendix E.4 for more details. We find that CRM outperforms Mixup across the board w.r.t. the worst group accuracy. That said, we would like to note that group DRO and logit shifting-based methods are considered strong baselines, which we had already compared within our work.
>
>
>  Dataset   | Method | Average Acc | Balanced Acc | Macro F1 | Worst Group Acc |
> |-----------|--------|-------------|--------------|----------|-----------------|
> | Waterbirds | ERM    | 77.9 (0.1)  | 75.3 (0.1)   | 83.2 (0.1) | 43.0 (0.2)     |
> |  | G-DRO  | 77.9 (0.9)  | 78.8 (0.7)   | 85.2 (0.8)| 42.3 (2.6)     |
> |  | Mixup  | 81.6 (0.1)  | 79.9 (0.1)   | 86.7 (0.1) | 52.2 (0.4)     |
> |  | CRM    | 87.1 (0.7)  | 87.8 (0.1)   | 93.2 (0.1) | 78.7 (1.0)     |
> | CelebA     | ERM    | 85.8 (0.3)  | 75.6 (0.1)   | 81.5 (0.1) | 39.0 (0.3)     |
> |      | G-DRO  | 89.2 (0.5)  | 86.8 (0.1)   | 92.5 (0.1) | 67.8 (0.8)|
> |      | Mixup  | 84.9 (0.2)  | 77.9 (0.2)   | 84.4 (0.3) | 42.8 (0.9)     |
> |      | CRM    | 91.1 (0.2)  | 89.2 (0.0)   | 94.2 (0.1) | 81.8 (0.5)     |
> | MetaShift  | ERM    | 85.7 (0.4)  | 81.7 (0.3)   | 88.8 (0.2) | 60.5 (0.5)     |
> |   | G-DRO  | 86.0 (0.3)  | 82.6 (0.2)   | 89.6 (0.2) | 63.8 (1.1)     |
> |   | Mixup  | 86.8 (0.0)  | 82.8 (0.1)   | 89.6 (0.1) | 62.8 (0.7)     |
> |   | CRM    | 87.6 (0.3)  | 84.7 (0.2)   | 91.4 (0.1) | 73.4 (0.4)     |
> | NICO++     | ERM    | 85.0 (0.0)  | 85.0 (0.0)   | 91.3 (0.3) | 35.3 (2.3)     |
> |      | G-DRO  | 84.0 (0.0)  | 83.7 (0.3)   | 91.0 (0.0) | 36.7 (0.7)     |
> |      | Mixup  | 85.0 (0.0)  | 84.7 (0.3)   | 91.0 (0.0) | 33.0 (0.0)     |
> |      | CRM    | 84.7 (0.3)  | 84.7 (0.3)   | 91.0 (0.0) | 40.3 (4.3)     |
>
> &nbsp;
>
> ## Why only report the average rankings of different methods?
>
> Our paper already has comprehensive results per dataset; we reported the average ranking of methods just to offer a concise summary. The detailed evidence (Table 1, 14, and others) in our paper overwhelmingly supports that **under compositional shifts CRM systematically outperforms all other baselines across all the datasets w.r.t worst group accuracy**, which is the metric of choice for robustness under spurious correlations.
>
> For comprehensive results we point you to the main results in Table 1, the ablations of CRM in Table 10, results on the original benchmark without dropping groups in Table 11, the multi-attribute experiment in Table 13, and results with additional baselines and Macro F1 score in Table 14. Further, we have provided results conditioned on both the dataset and the discarded group scenario for that dataset, please check Appendix D.1, Tables 5-9.

---

> ### Author Response · Authors · 2024-11-28
> **Request for Response**
>
> Please let us know if our response above has clarified your unresolved concerns, we have benchmarked CRM against Mixup (Table 16, Appendix E.4) and clarified your concern regarding average rankings as well. Hence, we believe we have thoroughly addressed both of your concerns, and we kindly ask you to consider revisiting your score.

---

> ### Author Response · Authors · 2024-12-02
> **Request for Response**
>
> As the rebuttal period is now coming to an end, we wish to remind the reviewer to please check our response above. We believe we have addressed both of your unresolved concerns. Regarding the Mixup baseline, we have also benchmarked it on MultiNLI and CivilComments, where our finding remains consistent: CRM outperforms Mixup, especially regarding the worst group accuracy. We provide these results below, and for the reader's convenience, we state results on other benchmarks again.
>
>  Dataset   | Method | Average Acc | Balanced Acc | Macro F1 | Worst Group Acc |
> |-----------|--------|-------------|--------------|----------|-----------------|
> | Waterbirds | ERM    | 77.9 (0.1)  | 75.3 (0.1)   | 83.2 (0.1) | 43.0 (0.2)     |
> |  | G-DRO  | 77.9 (0.9)  | 78.8 (0.7)   | 85.2 (0.8)| 42.3 (2.6)     |
> |  | Mixup  | 81.6 (0.1)  | 79.9 (0.1)   | 86.7 (0.1) | 52.2 (0.4)     |
> |  | **CRM**    | **87.1 (0.7)**  | **87.8 (0.1)**   | **93.2 (0.1)** | **78.7 (1.0)**     |
> | CelebA     | ERM    | 85.8 (0.3)  | 75.6 (0.1)   | 81.5 (0.1) | 39.0 (0.3)     |
> |      | G-DRO  | 89.2 (0.5)  | 86.8 (0.1)   | 92.5 (0.1) | 67.8 (0.8)|
> |      | Mixup  | 84.9 (0.2)  | 77.9 (0.2)   | 84.4 (0.3) | 42.8 (0.9)     |
> |      | **CRM**    | **91.1 (0.2)**  | **89.2 (0.0)**   | **94.2 (0.1)** | **81.8 (0.5)**     |
> | MetaShift  | ERM    | 85.7 (0.4)  | 81.7 (0.3)   | 88.8 (0.2) | 60.5 (0.5)     |
> |   | G-DRO  | 86.0 (0.3)  | 82.6 (0.2)   | 89.6 (0.2) | 63.8 (1.1)     |
> |   | Mixup  | 86.8 (0.0)  | 82.8 (0.1)   | 89.6 (0.1) | 62.8 (0.7)     |
> |   | **CRM**    | **87.6 (0.3)**  | **84.7 (0.2)**   | **91.4 (0.1)** | **73.4 (0.4)**     |
> | NICO++     | ERM    | 85.0 (0.0)  | 85.0 (0.0)   | 91.3 (0.3) | 35.3 (2.3)     |
> |      | G-DRO  | 84.0 (0.0)  | 83.7 (0.3)   | 91.0 (0.0) | 36.7 (0.7)     |
> |      | Mixup  | 85.0 (0.0)  | 84.7 (0.3)   | 91.0 (0.0) | 33.0 (0.0)     |
> |      | **CRM**    | **84.7 (0.3)**  | **84.7 (0.3)**   | **91.0 (0.0)** | **40.3 (4.3)**     |
> | MultiNLI        | ERM     | 69.1 (0.7)            | 69.8 (0.2)       | 77.0 (0.2)             | 7.2 (0.7)           |
> |                 | G-DRO   | 70.4 (0.2)            | 73.7 (0.2)       | 81.7 (0.2)             | 34.3 (0.2)          |
> |                 | Mixup   | 70.2 (0.1)            | 69.7 (0.1)       | 77.7 (0.2)             | 14.6 (1.0)          |
> |                 | **CRM**     | **74.7 (0.7)**           | **76.2 (0.6)**       | **85.8 (0.4)**             | **57.7 (3.0)**          |
> | CivilComments   | ERM     | 80.4 (0.2)            | 78.4 (0.0)       | 87.5 (0.1)             | 55.9 (0.2)          |
> |                 | G-DRO   | 80.1 (0.1)            | 78.9 (0.0)       | 87.9 (0.0)             | 61.6 (0.5)          |
> |                 | Mixup   | 80.1 (0.1)            | 78.2 (0.0)       | 87.3 (0.1)             | 55.4 (0.6)          |
> |                 | **CRM**     | **83.7 (0.1)**            | **78.4 (0.0)**       | **87.8 (0.0)**             | **68.1 (0.5)**          |

---

> > ### Comment · Reviewer_x1T8 · 2024-12-03
> >
> > I would thank the reviewers for their response. Although I still have concerns about the experimental settings (even if there are three attributes, I don't think they're enough to represent the setting considered in this paper), I would raise my score.

---

> ### Author Response · Authors · 2024-12-04
> **Experiments with more number of attributes**
>
> We are very thankful of your response and score increase. To address your concern about the multi-attribute case experiment, we  we have further augmented the CelebA dataset to include a total of 5 attributes. In addition to the (Gender, EyeGlasses) spurious attributes in the 3-attribute CelebA dataset (Appendix E.1), we further include (Hat, Earrings) attributes to result in a total of 32 groups with 5 attributes.
>
> The results are presented below, where we find that CRM still outperforms the baselines for generalizing under compositional shifts, especially w.r.t the worst group accuracy.
>
>
> | Dataset         | Method            | Average Acc  | Balanced Acc  | Macro F1  | Worst Group Acc |
> |-----------------|-------------------|-------------------|--------------------|----------------|-----------------------|
> | CelebAMultiAttr | ERM               | 93.9 (0.0)        | 80.5 (0.4)         | 84.9 (0.3)     | 2.8 (1.3)             |
> | | G-DRO             | 86.8 (0.3)        | 90.2 (0.2)         | 94.4 (0.2)     | 53.0 (4.4)            |
> | | LC                | 88.1 (0.0)        | 91.1 (0.0)         | 95.0 (0.0)     | 65.5 (1.0)            |
> | | sLA               | 88.2 (0.1)        | 91.0 (0.1)         | 94.9 (0.0)     | 68.0 (0.6)            |
> | | IRM               | 86.4 (0.6)        | 84.3 (0.1)         | 89.6 (0.1)     | 28.3 (0.6)            |
> | | VREx              | 84.5 (0.1)        | 87.5 (0.1)         | 91.4 (0.2)     | 7.6 (2.6)             |
> | | Mixup             | 93.7 (0.0)        | 80.4 (0.1)         | 85.2 (0.0)     | 1.2 (0.3)             |
> | | **CRM**               | **85.3 (0.0)**        | **88.1 (0.1)**         | **93.6 (0.1)**     | **71.3 (0.5)**            |

---

### Author Response · Authors · 2024-11-20
**General Response**

We thank the reviewers for their thoughtful feedback and constructive questions. We are encouraged by their recognition of the significance, novelty, and technical rigor of our work.
- **Reviewer Rmiq** noted that the study is "rigorous and comprehensive" and appreciated the clarity of the contributions through the running example.
- **Reviewer MuZN** highlighted the theoretical grounding of our approach and its importance for generalization under compositional shifts.
-  **Reviewer Rmiq** and **Reviewer x1T8** emphasized the significance of exploring compositional shifts, with **Reviewer x1T8** describing our formulation as providing "a foundation for further research".

We address the concerns raised by each reviewer in detail as an individual response to each reviewer. We address the multiple attribute experiment here, which was a shared concern for both **Reviewer x1T8** and **Reviewer Rmiq**.

&nbsp;

## **Multiple Attribute Experiment**

We ran a new experiment using the CelebA dataset, by adding the attribute 'Eyeglasses' as another spurious attribute, leading to a total of *3* attributes. The results are shown below as well as in the newly added Table 13 in Appendix E.1, with more details regarding the setup.

| Method | Average Acc | Balanced Acc | Worst Group Acc | WGA (No Groups Dropped) |
|--------|-------------|--------------|-----------------|-------------------------|
| ERM    | 94.1 (0.1)  | 77.4 (0.3)   | 21.6 (0.6)      | 19.0 (0.0)              |
| G-DRO  | 91.2 (0.5)  | 87.3 (0.3)   | 61.0 (2.1)      | 66.7 (2.3)              |
| LC     | 92.8 (0.0)  | 89.8 (0.1)   | 75.0 (1.3)      | 77.0 (2.0)              |
| sLA    | 93.0 (0.1)  | 89.7 (0.1)   | 76.0 (0.7)      | 77.0 (4.0)              |
| **CRM**    | **92.3 (0.2)**  | **91.6 (0.0)**   | **84.0 (0.3)**      | **86.3 (1.2)**  |

As can be seen from the results, CRM is the best approach w.r.t the worst group accuracy as well as the balanced group accuracy. Note that, as we had mentioned in the paper (line 442), average accuracy is not the best indicator of generalization under spurious correlations, hence we should pay more attention to the other metrics.

---

### Author Response · Authors · 2024-11-25
**Summary of Changes to the Draft**

We summarize the new sections added to the paper to address the reviewer's comments. We believe we have thoroughly addressed all the concerns raised by the reviewers. With the rebuttal period nearing its end, we wanted to check if our responses have sufficiently addressed your concerns. If so, we would be grateful if you could update your score accordingly.

- **Appendix E.1**: Multi-Attribute Experiment.  (Reviewer x1T8, Rmiq)

&#8594; Table 13 presents results for benchmarking CRM in the case of multiple attributes, showing that CRM outperforms other approaches in terms of both worst-group accuracy and group-balanced accuracy.


- **Appendix E.2**: Additional Baselines & Macro F1 Score (Reviewer x1T8, MuZN)

&#8594; Table 14 provides results with additional baselines (IRM, VREx) and metrics (Macro F1 score), providing more evidence for  CRM's superior generalization under compositional shifts. To better summarize the results, Table 15 provides the aggregate rank obtained by each method. CRM obtains the best rank w.r.t balanced accuracy, macro F1, and worst group accuracy, and CRM with empirical test prior obtains the best rank w.r.t average accuracy.

- **Appendix E.3:** Sample Complexity Analysis as function of $d$ (Reviewer MuZN)

&#8594; Figure 6 presents the results of our analysis where we evaluate CRM's generalization capabilities as we discard more groups from the training dataset for different values of $d$.

We have also additional details in Appendix D.2 which provides results for the ablation of CRM w.r..t the choice of test prior and extrapolated bias.

---

### Author Response · Authors · 2024-12-04
**Final Author Response**

We thank all the reviewers for their time and engagement during the rebuttal phase! Majority of the reviewer's concerns were about the empirical contributions of our work, which we thoroughly addressed with additional experiments and clarifications.

- We benchmarked CRM against three new baselines (IRM, VREx, Mixup) and additional metrics (Macro F1), and found that CRM outperforms them in its ability to generalize under compositional shifts. (Appendix E.2, E.4)

- We showed the superior generalization of CRM over baselines in the multi-attribute case via experiments on the multi-attribute CelebA dataset comprising of 3 attributes.  (Appendix E.1)

To address the concerns of **Reviewer x1T8** regarding the multi-attribute experiment, we have further augmented the CelebA dataset to include a total of 5 attributes. We present results for the case of 5-attribute CelebA dataset below, where each metric is averaged over the 32 compositional shift scenarios of discarding a group from the training and validation set.


| Dataset         | Method            | Average Acc  | Balanced Acc  | Macro F1  | Worst Group Acc |
|-----------------|-------------------|-------------------|--------------------|----------------|-----------------------|
| CelebAMultiAttr | ERM               | 93.9 (0.0)        | 80.5 (0.4)         | 84.9 (0.3)     | 2.8 (1.3)             |
| | G-DRO             | 86.8 (0.3)        | 90.2 (0.2)         | 94.4 (0.2)     | 53.0 (4.4)            |
| | LC                | 88.1 (0.0)        | 91.1 (0.0)         | 95.0 (0.0)     | 65.5 (1.0)            |
| | sLA               | 88.2 (0.1)        | 91.0 (0.1)         | 94.9 (0.0)     | 68.0 (0.6)            |
| | IRM               | 86.4 (0.6)        | 84.3 (0.1)         | 89.6 (0.1)     | 28.3 (0.6)            |
| | VREx              | 84.5 (0.1)        | 87.5 (0.1)         | 91.4 (0.2)     | 7.6 (2.6)             |
| | Mixup             | 93.7 (0.0)        | 80.4 (0.1)         | 85.2 (0.0)     | 1.2 (0.3)             |
| | **CRM**               | **85.3 (0.0)**        | **88.1 (0.1)**         | **93.6 (0.1)**     | **71.3 (0.5)**            |


We find that CRM still outperforms the baselines for generalizing under compositional shifts, especially w.r.t the worst group accuracy. Hence, the gains with CRM are consistent as we increase the total number of attributes.

We would also like to restate the significance of our theoretical contribution. Our method is the first to offer provable guarantees for extrapolating under compositional shifts, which also stands the test of experimentation on real benchmark datasets, as CRM systematically outperforms all baselines under such shifts.

---

### Meta-Review · Area_Chair_kALH · 2024-12-18

**Metareview:**

This submission is a borderline case. It proposed a problem called "compositional shift" as well as its solution called "compositional risk minimization". After the rebuttal, all three reviewers gave a rating of 6 that is "marginally above the acceptance threshold". During the internal discussions, none of three positive reviewers showed up, even after I cued them, suggesting that they were not truly supportive. In particular, reviewer x1T8 mentioned in his or her post-rebuttal comment that "although I still have concerns about the experimental settings (even if there are three attributes, I don't think they're enough to represent the setting considered in this paper), I would raise my score." My personal feeling is that the motivation of this work may not be strong enough --- as the authors mentioned, the problem under consideration is a **challenging and extreme** form of subpopulation shift, so why is such a hard problem meaningful in practice? Therefore, I don't think we can accept it for publication at such a competitive venue unless the authors can justify their motivation.

**Additional Comments On Reviewer Discussion:**

During the internal discussions, none of three positive reviewers (whose ratings were all 6) showed up, even after I cued them, suggesting that they were not truly supportive.

---

### Decision · Program_Chairs · 2025-01-22

Reject